# Characterization of Key Odorants in Scallion Pancake and Investigation on Their Changes during Storage

**DOI:** 10.3390/molecules26247647

**Published:** 2021-12-17

**Authors:** Binshan Liu, Shiqi Xu, Zhizhong Dong, Yuping Liu, Xiaoming Wei, Danqing Shao

**Affiliations:** 1School of Light Industry, Beijing Technology & Business University, Beijing 100048, China; 1930201009@st.btbu.edu.cn; 2Beijing Key Laboratory of Flavor Chemistry, Beijing Technology & Business University, Beijing 100048, China; xushiqi_btbu@163.com; 3COFCO Nutrition and Health Research Institute, Beijing 102209, China; dongzz@cofco.com (Z.D.); weixiaoming@cofco.com (X.W.); shaodanqing@cofco.com (D.S.)

**Keywords:** scallion pancake, GC-O, GC-MS, key odorants, storage, odor-active value

## Abstract

To characterize key odorants in scallion pancake (SP), volatiles were extracted by solvent extraction-solvent assisted flavor evaporation. A total of 51 odor-active compounds were identified by gas chromatography-olfactometry (GC-O) and chromatography–mass spectrometry (GC-MS). (*Z*/*E*)-3,6-Diethyl-1,2,4,5-tetrathiane was detected for the first time in scallion food. Application of aroma extract dilution analysis to extracts showed maltol, methyl propyl disulfide, dipropyl disulfide and 2-pentylfuran had the highest flavor dilution (FD) factor of 4096. Twenty-three odorants with FD factors ≥ 8 were quantitated, and their odor active values (OAVs) were calculated. Ten compounds with OAVs ≥ 1 were determined as the key odorants; a recombinate model prepared from the key odorants, including (*E*,*E*)-2,4-decadienal, dimethyl trisulfide, methyl propyl disulfide, hexanal, dipropyl trisulfide, maltol, acetoin, 2-methylnaphthalene, 2-pentylfuran and 2(5*H*)-furanone, successfully simulated the overall aroma profile of SP. The changes in odorants during storage were investigated further. With increasing concentrations and OAVs during storage, hexanal became an off-flavor compound.

## 1. Introduction

Wheat is the most widely cultivated cereal in the world; Asia contributes the most wheat cultivated area and production [1]. Wheat is considered as a major staple food in Asia [2], and is made into all kinds of local delicacies among Asian countries, such as Chinese steamed bread, shortening breads, noodles, biscuits, etc. [3,4,5,6]. Scallion pancake (SP) is a Chinese traditional wheat-based food. Because of its pleasant odor, SP is very popular in China. Its ingredients include wheat flour, water, cooking oil, chopped scallion and kitchen salt, which are all readily available, and its production process is simple, so it has become a family staple food. SP is different from breads, because the dough used for making SP is not fermented. The unfermented dough is shaped into a pancake, then sprinkled with chopped scallion and cooking oil. The pancake is folded several times so that the chopped scallion and cooking oil are homogeneously distributed in the pancake. After being cooked, the SP are edible.

Although SP is well-known and popular in China, there are few reports on its production methods or on the volatile constituents of SP. At present, available research about wheat product flavor compounds mainly focus on the odorants of breads [7,8]. In the past decades, more than 300 volatiles were reported in wheat bread crusts and crumbs, which includes abundant aldehydes, alcohols, organic acid and heterocyclic compounds [9]. Recently, the key odorants in wheat bread were determined by solvent-assisted flavor evaporation (SAFE) isolation, stable isotope dilution assays (SIDA) and odor active value (OAV) concept. Among the 19 quantitated compounds, 3-methyl-1-butanol (2711 μg/kg), 2-phenylethanol (2715 μg/kg) and acetic acid (97,863 μg/kg) showed higher concentrations in wheat bread crumbs, and 2-phenylethanol (1712 μg/kg), maltol (7472 μg/kg) and acetic acid (154,199 μg/kg) showed higher contents in crusts [7,10].

Scallion is one of the raw materials of SP, and its aroma compounds have been studied in depth. Sulfur-containing compounds are the main group of volatiles in scallions [11]. The sulfides identified in distilled scallion oil can be grouped as alk(en)yl di- or tri-sulfides, alk(en)ylthioalkyl alk(en)yl di- or tri-sulfides, alkyl tetra- or penta-thiaalkanes or -alkene(s) and thiaheterocycles [11,12].

To our knowledge, the current research on the food quality of wheat-based food during storage focuses on textural quality, oxidative stability and volatile compounds [13,14,15,16,17,18], and the research on the change in aroma profile during long-term storage is still lacking. The key odorants in SP have not been identified; no research has been conducted on the key odorants during storage, either. Sensomics, also called molecular sensory science approach [19], is a standard method for selecting and identifying the valuable compounds, which contribute to the characteristic aroma profile of samples [20]. According to this concept, the aroma profile can be analyzed qualitatively and quantitatively. 

The aims of the present study are to (1) screen the potent odorants of SP by using aroma extract dilution analysis; (2) quantitate the potent odorants; (3) determine the key odorants on the basis of OAVs; (4) investigate on the changes of concentrations and OAVs of potent odorants during storage. 

## 2. Results and Discussion

### 2.1. Potent Odor Compounds of SP

A total of 53 odor-active regions were detected, and their FD factors were measured. The results are showed in Table 1. Fifty-one odorants identified in SP were classified into seven chemical categories, namely alcohols, aldehydes, ketones, lactones, organic acids, sulfur-containing compounds and others. Among them, 45 compounds were positively identified by four methods; six compounds were tentatively identified by three different identification methods because of the lack of reference standards; two compounds could not be identified because only the sulfury-smelling note could be sniffed, and no specific MS signal appeared at the corresponding retention time. Most of them had been reported in soybean oil [21,22], wheat flour products [23,24,25], and allium products [26,27]. 

Seven aliphatic alcohols (including 2-methyl-2-butanol, 3-penten-2-ol, pentanol, (E)-2-hexenol, hexanol, 1-octen-3-ol, (E)-2-octenol) and three aromatic alcohols (including 2-furanmethanol, benzyl alcohol, phenethyl alcohol) were identified in SP. Most of them contributed green, fresh and floral notes but showed a lower FD factor (≤4) except 2-methyl-2-butanol and 2-furanmethanol with the higher FD factors. The FD factor of 2-furanmethanol was 128; it showed a sweet note. 2-Furanmethanol could be formed from 2-furanmethanal via the Cannizzaro reaction [28]. Coincidentally, both 2-furanmethanol and 2-furanmethanal were identified in SP. 2-Methyl-2-butanol showed the highest FD factor (256) among alcohols, but it contributed pungent notes, causing an unpleasant sensation in the nose during GC-O analysis. 2-Methyl-2-butanol was reported as a volatile compound in Moroccan argan oil [29] and roasted garlic with oils [30].

Most of the alcohols and aldehydes detected in SP could be formed by oxidative breakdown of unsaturated fatty acids [31,32]. Ten aldehydes were found in SP. Hexanal (green), 3-methyl-2-butenal (fruity), vanillin (vanilla) showed FD factors of 64, 128, and 64, respectively. Both benzaldehyde (nutty, almond) and (*E*,*E*)-2,4-decadienal (fatty) exhibited a higher FD factor of 2048. The former was considered to be a product of Strecker degradation [33], and the later was generated through lipid oxidation [34]. Both of them had reported in fried shallot oil [26]. Furfural and 5-hydroxymethylfurfural which presented sweet notes were reported to be derived from Maillard reaction and caramelization reaction. Nonanal (soapy), (*E*)-2-octenal (cucumber-like), 2-formylpyrrole (musty), 5-hydroxymethylfurfural (buttery) showed such lower FD factors that could be detected only in first few dilutions.

Seven ketones were identified as odor-active compounds in SP. Four of them, namely, acetoin (sweet), cyclopentenone (roasted), 2(5*H*)-furanone (buttery) and maltol (sweet), showed a higher FD factor ranging from 32 to 4096. Acetoin, 2(5*H*)-furanone and maltol can be formed during Maillard reactions [35]. Furthermore, 2(5*H*)-furanone was determined as one of the most abundant volatiles in wheat muffins [36]. Acetol (caramel-like), 1-(2-furanyl)-2-hydroxyethanone (sweet), 2-pyrrolidinone (pungent) could only be smelled in undiluted extracts. 2-Pyrrolidinone might be from the soybean oil used, because it has been reported in tempeh [37], which is soybeans product.

Lactones and acids identified in SP had a weak aroma intensity, except γ-butyrolactone (creamy). Lactones and acids, such as γ-butyrolactone and low-carbon acids, could be produced by the thermal oxidative degradation of saturated fatty acids [27], because soybean oil was used as a SP material, and it was rich in fatty acids.

Eleven sulfur-containing compounds, including seven sulfides, two sulfones and two heterocycle compounds, were detected as odor-active compounds and had strong odor intensity. They should be from the scallions used in SP, because scallions contained many sulfur-containing compounds. Methyl propyl disulfide (onion-like) and dipropyl disulfide (alliaceous) displayed the highest FD factors of 4096. Other disulfides and trisulfides showed FD factors ranging from 64 to 512. All of them were reported in distilled scallion oil [11]. Dipropyl disulfide and dipropyl trisulfide had been identified as thermal degradation products of S-propylcysteine sulfoxide and cleavage products of S-propylcysteine [38]. To identify the four compounds (**43**, **44**, **46**, **47**), their reference substances were obtained by isolation of the by-products of dithiazine synthesis. Interestingly, there was no mention in the literature that the reaction could form **43** and **44**. *Z*-isomer and *E*-isomer had similar mass spectra data, so the retention indices were used to distinguish them. Although *Z*-isomer and *E*-isomer could be separated well on DB-Wax column, their LRIs on DB-Wax column had not been reported. However, their LRIs on the HP-5MS column could be obtained from Andriamahravo’s database [39]. LRIs on the HP-5MS column of four compounds were 1331/1339 for (*Z*/*E*)-1-propenyl propyl trisulfide and 1610/1613 for (*Z*/*E*)-3,6-diethyl-1,2,4,5-tetrathiane. *Z*-isomers of the two pairs of compounds had smaller LRIs than their respective *E*-structure, so their *Z*- and *E*- structures could be distinguished by HP-5MS column isolation. Figure 1C,D showed the GC-MS signals of the synthesized four compounds on HP-5MS column. The peak areas of (*Z*)-1-propenyl propyl trisulfide and (*Z*)-3,6-diethyl-1,2,4,5-tetrathiane were slightly smaller than those of their trans structures. According to the abovementioned results, it could be deduced that the smaller areas were *Z*-isomers, and the bigger areas were *E*-isomers in Figure 1A,B obtained on DB-Wax column. According the results above, **43**, **44**, **46** and **47** were identified in SP. (*Z*/*E*)-3,6-Diethyl-1,2,4,5-tetrathiane (onion-like, sweet) was once reported in cooked white and red onions, leek, shallots, chives, and spring onions [40], but it was first identified in scallion-containing foods. Both dimethyl sulfone (burnt) and dimethyl sulfoxide (oily) showed higher FD factors with 512 and 256, respectively. Moreover, dimethyl sulfone had been identified in white bread [24]; both were found in rapeseed oil [41]. Notably, allyl sulfides were absent in this study, and the result was consistent with Eric Block’s conclusion that allyl-group-containing compounds do not exist in scallion [42].

Pyrazine, 2-pentylfuran, naphthalene, 2-naphthalene and two unknown compounds were also detected in SP. 2-Penthylfuran (green, earthy) with an FD factor of 4096 might be produced from the pyrolysis of carbohydrates [43]. Naphthalene and 2-methylnaphthalene were reported in steamed bread [8], and had the FD factors of 4 and 64 in SP, respectively. Two unknowns were presumed to sulfur-containing compounds because they had sulfury note.

### 2.2. Quantitative Analysis and Determination of Key Odorants by OAVs

A total of 23 compounds with FD ≥ 8 were quantitated, with the results listed in Table 2. They had a wide range of concentrations. Maltol (5073.26 μg/kg) showed the highest level in SP; it was reported that maltol was also presented in wheat bread crust at higher concentrations (7472 μg/kg) [4]. 2-Furanmethanol (1119 μg/kg), 2(5*H*)-furanone (1642.24 μg/kg) and dihydro-4-hydroxy-2(3*H*)-furanone (1244.31 μg/kg) had higher contents, too. Only the concentrations of these four compounds above were at ppm level. Furfural, benzaldehyde, (*E*,*E*)-2,4-decadienal showed close concentrations from 131.79 μg/kg to 164.46 μg/kg. 3-Methyl-2-butenal (13.37 μg/kg) and vanillin (34.72 μg/kg) showed at lower concentrations. Acetoin and cyclopentenone present a concentration at 177.9 μg/kg and 67.68 μg/kg, respectively. Among the sulfur-containing compounds, four sulfides, including methyl propyl disulfide (28.9 μg/kg), dipropyl disulfide (35.56 μg/kg), dimethyl trisulfide (29.77 μg/kg), dipropyl trisulfide (9.92 μg/kg), possessed lower contents. However, the concentrations of dimethyl sulfoxide (264.31 μg/kg) and dimethyl sulfone (120.34 μg/kg) were several times or even tens higher than those of four sulfides.

OAVs is the concept about aroma contribution combining quantification results with human olfactory senses to flavor compounds. By calculating whether the OAVs were ≥ 1, key aroma compounds would be determined, and volatiles with little aromatic contribution would be screened out. The calculations of OAVs to quantitated compounds were performed, and the data were listed in Table 3. (*E*,*E*)-2,4-Decadienal (fatty) presented the highest OAV of 6091, followed by dimethyl trisulfide with sauce-like notes (OAV = 3007), methyl propyl disulfide (onion-like, OAV=270), hexanal (green, OAV = 230), dipropyl trisulfide (garlic-like, OAV = 52), 2-pentylfuran (green, OAV = 3). Maltol and 2(5*H*)-furanone had higher concentrations in SP, but their OAVs were only 24 and 2, respectively. Meanwhile, acetoin had an OAV of 13, and 2-methylnaphthalene was with an OAV = 6. Ten compounds with OAV > 1 above were considered as key odorants of SP.

### 2.3. Recombination

To get an impression on the overall odor profile of SP, a descriptive sensory analysis was performed, and six notes (sauce-like, fatty, sulfurous, sweet, roasty, green) were determined and scored; the results obtained were plotted in a spider diagram (Figure 2). It can be seen that the fatty note got the highest score with 6.5, followed by the roasty note with 5.7, and the sauce-like note with 5.4. This was consistent with people’s impression of Chinese style pancake flavor. Sulfurous attribute got a medium score (4.6), because only few chopped scallions were used. Most panelists gave lower scores on sweet and green notes (3.9 and 3.6, respectively). The odorants with these six notes should be paid more attention in the following analyses.

To confirm whether the key odorants identified contributed greatly to the overall odor profile of SP, an odor recombination experiment was conducted by mixing the key odorants together at the concentrations determined by quantitation analyses. The odor characteristics of the recombinate and SP were evaluated by scoring the six attributes; the results obtained are shown in the spider diagram in Figure 2. It can be seen that the SP sample had the same score in fatty note as the reconstitution model. In the recombinate, only (*E*,*E*)-2,4-decadienal had a fatty note, so it was regarded as the important contributor to the fatty attribute in SP. The recombinate and SP had close scores in terms of the roasty note. There were some slight differences in their green and sulfury notes; maybe the interactions between volatile compounds or between volatiles and matrix influenced their odor intensities. However, the SP sample possessed the higher aroma intensities in sauce-like and sweet notes than the reconstituted SP; the reason might be that the recombinate lacked certain sulfur-containing compounds which had not been identified or could not be available on market, such as (*Z*/*E*)-3,6-Diethyl-1,2,4,5-tetrathiane had sweet and salty notes, but they were not isolated completely in synthesis section, and the two unidentified compounds also had some sweet and salty notes.

### 2.4. Changes of Aroma Compounds during Storage

Table 4 showed the changes in concentration of potent odor-active compounds during storage, and the concentration data with * meant OAVs ≥ 1. There were some differences among the appearance of samples because of their non-uniformity, but the data obtained could still show a general changing tendency. From month 1 to month 14, the contents of these odorants underwent varying degrees of change. The concentrations of **8**, **15**, **17**, **24**, **30**, **49**, and **53** fluctuated; **1** presented a decreasing concentration. However, for **11**, **12**, **16**, **20**, **21**, **23**, **25**, **28**, **37**, **38**, **39**, **41**, **42**, **45**, **48**, their levels generally increased. Among them, the concentrations of **11**, **23**, **38**, and **42** rose dramatically from month 1 to month 14 (**11**, 553.03 to 24,307.34 μg/kg; **23**, 67.68 to 403.15 μg/kg; **38**, 35.56 to 282.7 μg/kg; **42**, 9.92 to 99.08 μg/kg). The soaring of hexanal (**11**) concentration may be mainly caused by the oxidative breakdown of unsaturated fatty acids during storage. The content of vanillin (sweet, vanilla) exceeded its detection threshold at month 3. The concentrations of dipropyl disulfide (sulfury, alliaceous) and dimethyl sulfoxide (oily) ascended and exceeded their respective thresholds at month 12, but it was reported that their contents in fried shallot oils decreased during storage [22]. Maybe more sulfur-containing compounds were released or formed during SP storage, and the matrix stopped the weakening of sulfur-containing compounds, which led to an increase in sulfur-containing abundance. Hexanal had the most significant change, and its concentration increased by tens of times its original concentration. This was mainly attributed to lipid oxidation during storage. 

OAV represents the contribution of a single compound to the overall aroma profile of the sample, which could be used to compare the aroma profile difference between the stored sample (month 3, 10, 12, 14) and the original sample (month 1). Table 5 focuses on the changes in 13 odorants with OAV ≥ 1, where **20**, **38**, **41** were absent in the original sample. In general, no odors with new attributes were produced during storage, but the overall aroma profile changed due to the significant changes in OAV. The total OAV first increased slightly, then decreased in month 10, and finally experienced a sharp increase until month 14. (*E*,*E*)-2,4-Decadienal and dimethyl trisulfide, which had similar trends to the total OAV, accounted for the majority of total OAV in the first ten storage months, so the decrease of their OAV led to the decrease of total OAV in month 10. Hexanal (OAV was 230 in month 1 and 10,128 in month 14) and dimethyl trisulfide (OAV was 3007 in month 1 and 7012 in month 14) were the main cause for the increase of OAV. However, because dimethyl trisulfide has a high OAV in the original sample, its final OAV was only doubled; the OAV of hexanal in month 14 had increased to about 500 times than in month 1, and was higher than dimethyl trisulfide dominating the total OAV, which meant it had become a major contributor to the aroma profile of SP. The formation of more hexanal resulted in the change in SP flavor. The increasing concentrations of hexanal had a negative influence on SP flavor and became an off-flavor compound. Based on the results obtained, hexanal could be regarded as an indicator of off-flavor.

## 3. Materials and Methods

### 3.1. Materials

SP was obtained from COFCO Nutrition and Health Research Institute and individually packaged in airtight plastic boxes. The main ingredients of SP were wheat flour, drinking water, modified starch, soybean oil, scallion and kitchen salt. Before the experiment, all samples were stored at room temperature and under no-light conditions.

### 3.2. Chemicals

For the purpose of qualitative and quantitative analyses of aroma-active compounds, the following reference compounds were purchased from commercial sources given in parentheses: **1**, **3**, **5**, **9**–**11**, **24**, **36**, **42**, **49** (Macklin Biochemical Co., Ltd., Shanghai, China); **6**, **22** (Adamas Reagent Co., Ltd., Shanghai, China); **7**, **15**, **17**, **19**, **27**, **41**, **48** (Aladdin Reagents Co., Ltd., Shanghai, China); **8**, **12**, **23**, **25**, **28**, **29**, **37**–**39**, **45**, **53** (J&K Chemical Ltd., Beijing, China); **16**, **32**, **33** (Beijing Peking University Zoteq Co., Ltd., Beijing, China); **30** (Alfa Aesar reagent company, Shanghai, China); **31** (Beijing MREDA Technology Co., Ltd., Beijing, China). Compounds **43**, **44**, **46**, **47** were synthesized according to Krafft’s method with some modifications [23] and the detailed synthesis steps are shown in the Appendix A; chemicals for synthesis were supplied as followed: propanal, (NH_4_)_2_S, tetrahydrofuran (THF), methyl t-butyl ether (t-BuOMe) (Aladdin Reagents Co., Ltd., Shanghai, China); dichloromethane, diethyl ether and hydrochloric acid (Sinopharm Chemical Reagent Co., Ltd., Beijing, China); Na_2_SO_4_ (anhydrous), NaBrO_3_, NaBr, NaHCO_3_, NaS_2_O_3_, n-hexane (Macklin Biochemical Co., Ltd., Shanghai, China). N-alkane standards (C_7_–C_28_) were purchased from Sigma-Aldrich Chemical Co. Liquid nitrogen and ultra-high-purity (UHP) helium were supplied by Beijing RUIZHX Technology Co., Ltd. (Beijing, China).

### 3.3. Isolation of Volatiles from SP by Direct Solvent Extraction−Solvent Assisted Flavor Evaporation (DSE-SAFE)

SP powder was obtained by removing brown and burnt crusts, dividing the crumbs into 1 cm^3^ cubes, freezing in liquid nitrogen and grinding with a commercial blender. The powder (50 g) was extracted by dichloromethane (100 mL × 2) under vigorous stirring. The filtrates were combined. Volatiles were isolated by a SAFE apparatus (Shenxian Jingxing Glassware Co., Ltd., Shenxian, China) under 2.5 × 10^−5^ mbar (Edwards TIC Pumping Station from BOC Edwards, Crawley, UK). The distillate was dried with anhydrous Na_2_SO_4_ overnight, concentrated to about 5 mL using a Vigreux column (15 cm) at 47 °C and further concentrated to 1 mL using a cool nitrogen stream. The final extract was transferred to a 2-mL GC glass vial sealed with PTFE/silicone pad and stored at −40 °C until further analyses.

### 3.4. Analysis of Aroma-Active Compounds in SP

#### 3.4.1. GC-MS Analysis

The identification and quantification of aroma-active compounds were carried out using an Agilent 7890B GC series connected with a 5977A mass selective detector (Agilent Technologies, Santa Clara, CA, USA). The concentrated distillate (1 μL) from SAFE was injected into the injection port; the splitless mode was used.

The operating parameters of the instrument were set as follows: the temperatures of GC injection port, ion source, and quadrupole were set at 240 °C, 230 °C, and 150 °C, respectively. Mass spectra was visualized in election ionization mode with a 70 eV ionization energy. MS data were acquired in full-scan mode with the mass range of 33–350 amu. For SAFE isolate, the oven temperature was held at 40 °C for 5 min, raised to 230 °C as final temperature at a ramp rate of 6 °C/min and held for 15 min. The separation of volatile compounds was performed in both a HP-5MS capillary column (30 m × 0.25 mm × 0.25 μm) and a DB-WAX capillary column (30 m × 0.25 mm × 0.25 μm). The constant flow rate of helium in both capillary columns was 1 mL/min.

#### 3.4.2. GC-FID-O Analysis

SP extracts were analyzed in an Agilent 7890B GC equipped with a flame ionization detector (Agilent Technologies, Santa Clara, CA, USA) and an olfactometer (ODP3 Gerstel GmbH, Mülheim an der Ruhr, Germany). Concentrate (1 μL) was injected on column and recorded the chromatographic signal at 280 °C. Other chromatographic parameters were the same as GC-MS. One DB-WAX capillary column (30 m × 0.25 mm × 0.25 μm) was used to separate the volatile compounds, and the effluent flowing from the capillary column would enter FID and sniffer in a 1:2 ratio by two deactivated capillaries. The ODP transfer line and sniffing port were heated at 240 °C and 120 °C, respectively. Humidified air would flow out together with the separated compounds to protect the panelists’ nose and avoid olfactory fatigue. To ensure the accuracy and precision of the experiments, the sniffing experiments were carried out by three well-trained flavor chemists including two females and one male. The odor descriptor and retention time of odor-active substances sniffed were recorded throughout the GC-FID-O experiments. When a substance was detected by at least two panelists, it was considered as an odor-active compound.

#### 3.4.3. Identification of Aroma-Active Compound 

A series of n-alkanes was analyzed by GC-O and GC-MS, and their retention times were used for calculating linear retention indices (LRIs) by the equation of Van Den Dool and Kratz [46] under the conditions corresponding to the above experiments. An odor-active compound would be identified positively by comparing its aroma characteristic, MS data, and retention index with those of the reference compound if the reference compound was available. When reference compounds were difficult to obtain, unknown substances were identified tentatively by comparing odor characteristic, MS data, retention indices with NIST14.L mass spectral library, online databases (NIST Chemistry WebBook, www.thegoodscentscompany.com, (accessed on 10 December 2021)) as well as the corresponding data reported in the published literatures.

#### 3.4.4. Aroma Extract Dilution Analysis (AEDA)

The flavor dilution (FD) factor of every odor-active compound was determined by AEDA. The isolate obtained by SAFE was stepwise diluted (2, 4, 8, etc.) by the addition of dichloromethane, and the diluted isolate was analyzed by GC-O until the odor of every odor-active compound was unperceivable at the sniffing port. Each diluted isolate was sniffed three times. FD factor of the odorant was defined as the highest perceivable dilution ratio.

#### 3.4.5. Quantitation of Odor-Active Compounds 

The odorants with an FD factor ≥ 8 and corresponding standard reference available were quantitated by internal standard curve method. 3-Octanol (internal standard, 300 μL, 10^−4^ g/mL in dichloromethane) were spiked into the SP powder prior to extraction experiment; the volatile isolate was obtained according to the method mentioned above. Seven different concentration stock solutions composed of 3-octanol and quantitated compounds were prepared and analyzed by GC-MS under the conditions mentioned above, except that selected ion monitoring mode was used. The standard curves forced through the origin were built, where y represented the peak area ratio of the odorant to 3-octanol and x represented the concentration ratio of the odorant to 3-octanol. The experiment was replicated in triplicate.

### 3.5. Quantitative Descriptive Analysis (QDA)

The SP aroma profile was analyzed by a well-trained panel of 16 members (eight males and eight females, ages 21–27 years) from Beijing Technology and Business University. The SP crumbs cut into small pieces were presented to panelists in the preliminary experiment, and six notes (sauce-like, fatty, sulfurous, sweet, roasty, green) were determined. For QDA, two sessions were performed. In the first session, the panelists were trained to distinguish six note attributes, and each of them was presented by one specific odorant solution, prepared by 50-fold to detective threshold in water, namely, sauce-like (dimethyl trisulfide), fatty (*E*,*E*-2,4-decdienal), sulfury (dipropyl trisulfide), sweet (maltol), roasty (pyrazine), and green (hexanal). In the second session, the panelists were asked to score the intensity of each note on a scale of 0 to 9 integers, in which 0 meant odorless and 9 meant very strong. The results were plotted in a spider web.

### 3.6. Measurement of Odor Detection Threshold in Water and Calculation of Odor Active Value (OAV)

To evaluate the contribution of a single aroma compound to the overall aroma profile, the concept of OAV was applied based on quantitative results and odor threshold values. To calculate OAVs, the detection thresholds of **24**, **30**, **37**, **41**, **42**, **45** were measured according to the Michael Czerny method with some modification [44]. The modification point was that the solutions were prepared with propylene glycol and water instead of ethanol and water. OAVs were calculated as a ratio of their concentrations to odor detection thresholds [44,45].

### 3.7. Aroma Recombination

Based on quantitative results, a model containing 10 compounds with OAVs ≥ 1 were prepared in water at their concentrations in SP. The panelists were asked to score the recombinate as previously described in *Quantitative Descriptive Analysis.*

### 3.8. Determination of Changes of Odorant Concentrations and OAVs during Storage

To investigate the changes in odorant concentrations and OAVs during storage, SP was sampled and analyzed on month 1, 3, 10, 12 and 14.

### 3.9. Statistical Analysis

Quantitative experiments were carried out in triplicate and results were displayed as means ± standard deviation (SD). Statistical analyses were performed by one-way ANOVA combined with Duncan’s multiple-range test (*p* < 0.05) using the SPSS 23 (SPSS Inc., Chicago, IL, USA) software.

## 4. Conclusions

In summary, 53 odor-active regions were detected, and 10 odorants were determined as key odorants in SP by GC-MS, GC-O, quantitative experiments and OAVs. The odorants of SP were associated with its main raw materials (wheat flour, scallion and soybean oil). The concept of OAV combined with orthonasal and concentration of odorant was also used to characterize the aroma changes of SP during storage. Hexanal was considered as an off-flavor compound, with its dramatic increase in concentration and emergence as a dominant aroma profile contributor during storage. The results of this study can provide reference for determining the shelf life of SP or other staple foods rich in oil in Asia. Further study about the mechanisms involving odorant formation and changes should be carried out, and the formation of hexanal should be controlled during storage, which is conducive to the development of food storage technology.

## Figures and Tables

**Figure 1 molecules-26-07647-f001:**
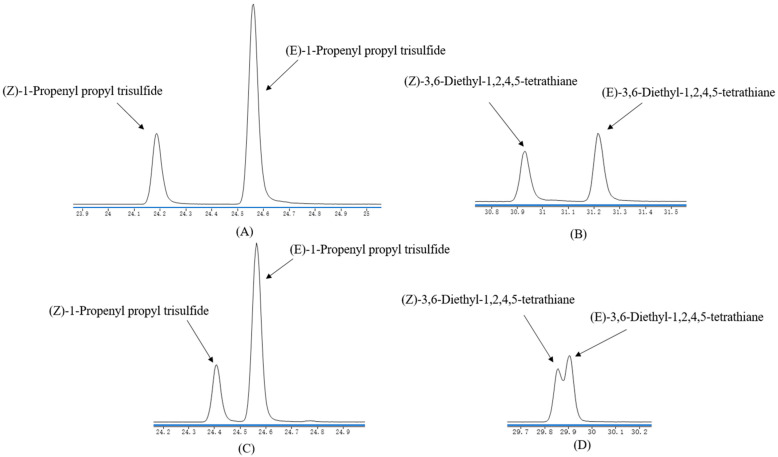
GC-MS signals of synthesized **43, 44, 46, 47** by DB-WAX (**A**,**B**) and HP-5MS (**C**,**D**) column separation.

**Figure 2 molecules-26-07647-f002:**
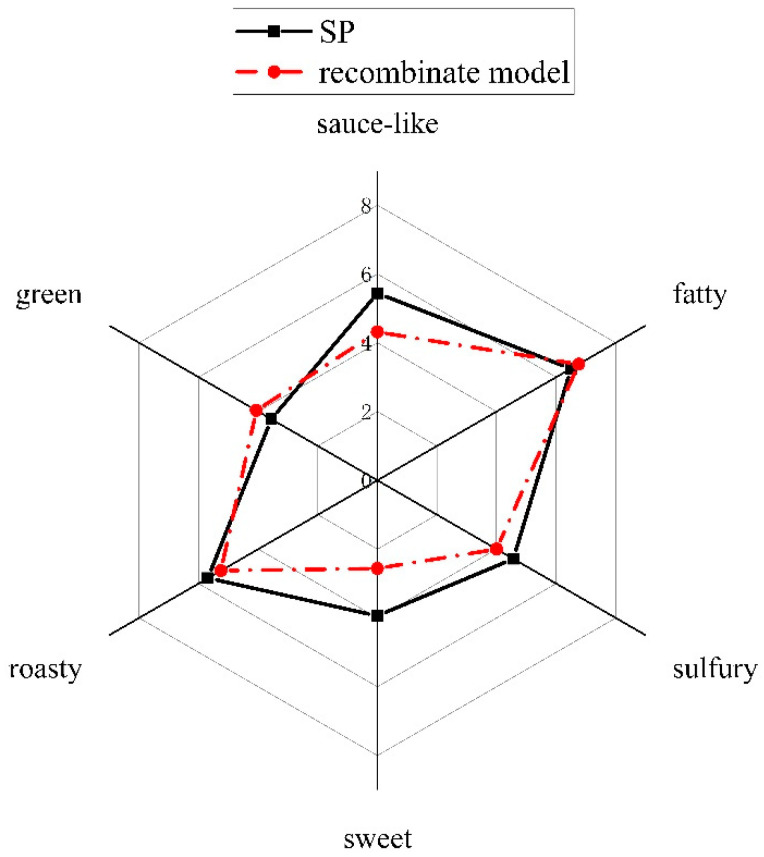
Quantitative descriptive analysis between SP and recombinate model.

**Table 1 molecules-26-07647-t001:** Aroma-active compounds identified by GC-O and GC-MS from SP.

No.	Compounds	LRI		Odor Descriptions ^c^	FD ^d^	Identification ^e^
		DB-WAX ^a^	HP-5MS ^b^			
**Alcohols**						
**1**	2-methyl-2-butanol	1008	- ^f^	pungent	256	MS, RI, O, S
**2**	3-penten-2-ol	1179	782	green	2	MS, RI, O
**3**	pentanol	1259	771	fusel, sweet	2	MS, RI, O, S
**4**	(*E*)-2-hexenol	1294	859	green, fresh	˂2 ^g^	MS, RI, O
**5**	hexanol	1362	868	green, fruity	˂2	MS, RI, O, S
**6**	1-octen-3-ol	1459	983	mushroom, green	4	MS, RI, O, S
**7**	(*E*)-2-octenol	1626	1062	green, citrus	2	MS, RI, O, S
**8**	2-furanmethanol	1670	863	sweet, caramellic	128	MS, RI, O, S
**9**	benzyl alcohol	1892	1041	floral, rose	˂2	MS, RI, O, S
**10**	phenethyl alcohol	1932	1122	floral, rose	4	MS, RI, O, S
**Aldehydes**						
**11**	hexanal	1093	805	green, aldehyde-like	64	MS, RI, O, S
**12**	3-methyl-2-butenal	1202	787	fruity, sweet	128	MS, RI, O, S
**13**	nonanal	1401	1105	aldehyde, soapy	˂2	MS, RI, O, S
**14**	(*E*)-2-octenal	1439	1057	fresh, cucumber-like	4	MS, RI, O, S
**15**	furfural	1470	838	sweet, roasted	8	MS, RI, O, S
**16**	benzaldehyde	1538	962	nutty, almond	2048	MS, RI, O, S
**17**	(*E*,*E*)-2,4-decadienal	1827	1316	fatty	2048	MS, RI, O, S
**18**	2-formylpyrrole	2044	-	musty, beefy	˂2	MS, RI, O, S
**19**	5-hydroxymethylfurfural	2519	1231	fatty, buttery	4	MS, RI, O, S
**20**	vanillin	2588	1399	sweet, vanilla	64	MS, RI, O, S
**Ketones**						
**21**	acetoin	1289	716	sweet, creamy	64	MS, RI, O, S
**22**	acetol	1302	-	sweet, caramellic	˂2	MS, RI, O, S
**23**	cyclopentenone	1364	805	roasted	512	MS, RI, O, S
**24**	2(5*H*)-furanone	1771	921	buttery	32	MS, RI, O, S
**25**	maltol	1988	1116	sweet, buttery	4096	MS, RI, O, S
**26**	1-(2-furanyl)-2-hydroxyethanone	2027	1082	sweet	˂2	MS, RI, O
**27**	2-pyrrolidinone	2064	-	pungent	˂2	MS, RI, O, S
**Lactones**						
**28**	γ-butyrolactone	1643	913	creamy, caramellic	1024	MS, RI, O, S
**29**	(±)-pantolactone	2048	1038	cotton	˂2	MS, RI, O, S
**30**	dihydro-4-hydroxy-2(3*H*)-furanone	2619	1173	sweet, creamy	32	MS, RI, O, S
**Acids**						
**31**	acetic acid	1454	-	sour	˂2	MS, RI, O, S
**32**	pentanoic acid	1755	890	sour, sweaty	2	MS, RI, O, S
**33**	hexanoic acid	1858	1022	sour, fatty	2	MS, RI, O, S
**34**	heptanoic acid	1976	1086	rancid, cheese	˂2	MS, RI, O
**35**	octanoic acid	2088	1172	fatty, waxy	2	MS, RI, O, S
**36**	nonanoic acid	2197	1287	waxy, cheese	2	MS, RI, O, S
**Sulfides**						
**37**	methyl propyl disulfide	1240	933	sulfury, onion-like	4096	MS, RI, O, S
**38**	dipropyl disulfide	1389	1107	sulfury, alliaceous	4096	MS, RI, O, S
**39**	dimethyl trisulfide	1390	968	sulfury, sauce-like	256	MS, RI, O, S
**40**	methyl propyl trisulfide	1544	1151	sulfury, garlic-like	128	MS, RI, O
**41**	dimethyl sulfoxide	1590	852	oily	256	MS, RI, O, S
**42**	dipropyl trisulfide	1689	1329	sulfury, garlic-like	512	MS, RI, O, S
**43**	(*Z*)-1-propenyl propyl trisulfide	1797	1336	sulfury, onion-like	64	MS, RI, O, S
**44**	(*E*)-1-propenyl propyl trisulfide	1818	1344	sulfury, onion-like	128	MS, RI, O, S
**45**	dimethyl sulfone	1914	926	sulfury, burnt	512	MS, RI, O, S
**46**	(*Z*)-3,6-diethyl-1,2,4,5-tetrathiane	2217	1587	sulfury, onion-like, sweet, salty	32	MS, RI, O, S
**47**	(*E*)-3,6-diethyl-1,2,4,5-tetrathiane	2235	1592	sulfury, onion-like, sweet, salty	64	MS, RI, O, S
**Others**						
**48**	pyrazine	1215	731	sweet, roasted	128	MS, RI, O, S
**49**	2-pentylfuran	1240	996	green, earthy	4096	MS, RI, O, S
**50**	unknown	1424	-	sulfury, onion-like	128	-
**51**	unknown	1456	-	sulfury	256	-
**52**	naphthalene	1760	1178	pungent, tarry	4	MS, RI, O
**53**	2-methylnaphthalene	1874	1311	floral, woody	64	MS, RI, O, S

^a^ Retention index on an DB-WAX column. ^b^ Retention index on an HP-5MS column. ^c^ detected by GC-O. ^d^ Flavor dilution factors. ^e^ Identification methods: MS, mass spectrum comparisons with data in NIST14 library. RI, comparing linear retention indices (LRI) on two columns (DB-WAX and HP-5MS) with those in the literature; O, confirmed by odor descriptions; S, confirmed by standard reference. ^f^ Not being detected. ^g^ Only can be detected at undiluted concentrations.

**Table 2 molecules-26-07647-t002:** Concentrations of 23 aroma-active compounds in SP.

No.	Compounds	Quantitative ions (*m*/*z*)	Slope ^a^	R^2^	Concentration (μg/kg) ^b^
**1**	2-methyl-2-butanol	73	0.6120	0.997	12.41 ± 0.07
**8**	2-furanmethanol	98	0.3591	0.995	1119 ± 8.09
**11**	hexanal	72	0.0592	0.992	553.03 ± 39.16
**12**	3-methyl-2-butenal	83	0.2151	0.998	13.37 ± 1.04
**15**	furfural	96	0.7631	0.994	144.13 ± 0.05
**16**	benzaldehyde	106	0.9063	0.999	131.79 ± 0.29
**17**	(*E*,*E*)-2,4-decadienal	152	0.0911	1.000	164.46 ± 0.37
**20**	vanillin	152	1.3559	0.998	34.72 ± 0.32
**21**	acetoin	88	0.1120	1.000	177.9 ± 1.76
**23**	cyclopentenone	82	0.7047	0.998	67.68 ± 0.7
**24**	2(5*H*)-furanone	84	0.2358	0.997	1642.24 ± 62.49
**25**	maltol	126	0.5552	0.995	5073.26 ± 20.19
**28**	*γ*-butyrolactone	86	0.1726	0.998	217.78 ± 2.77
**30**	dihydro-4-hydroxy-2(3*H*)-furanone	102	0.0623	0.997	1244.31 ± 20.75
**37**	methyl propyl disulfide	122	0.2886	0.995	28.9 ± 0.14
**38**	dipropyl disulfide	150	0.8402	0.996	35.56 ± 0.19
**39**	dimethyl trisulfide	126	1.1960	0.991	29.77 ± 0.26
**41**	dimethyl sulfoxide	78	0.5521	0.996	264.31 ± 2.5
**42**	dipropyl trisulfide	182	1.1046	0.990	9.92 ± 0.14
**45**	dimethyl sulfone	94	0.5662	0.994	120.34 ± 0.71
**48**	pyrazine	80	1.2723	0.995	41.12 ± 0.1
**49**	2-pentylfuran	138	0.4601	0.998	19.65 ± 0.06
**53**	2-methylnaphthalene	142	1.4490	0.992	16.93 ± 0.02

^a^ Slope of standard curve which x represented the concentration ratio of volatile standard to internal standard and y represented the peak area ratio of volatile standard to internal standard. ^b^ Concentration expressed as average concentration ± standard deviation (*n* = 3).

**Table 3 molecules-26-07647-t003:** OAVs of aroma-active compounds in SP.

No.	Compound	Concentration (μg/kg)	DOT ^a^	OAV
**17**	(*E*,*E*)-2,4-decadienal	164.46	0.027 ^b^	6091
**39**	dimethyl trisulfide	29.77	0.0099 ^b^	3007
**37**	methyl propyl disulfide	28.90	0.107 ^c^	270
**11**	hexanal	553.03	2.4 ^b^	230
**42**	dipropyl trisulfide	9.92	0.19 ^c^	52
**25**	maltol	5073.26	210 ^d^	24
**21**	acetoin	177.90	14 ^d^	13
**53**	2-methylnaphthalene	16.93	3 ^d^	6
**49**	2-pentylfuran	19.65	5.8 ^d^	3
**24**	2(5*H*)-furanone	1642.24	714 ^c^	2
**20**	vanillin	34.72	53 ^d^	˂1
**41**	dimethyl sulfoxide	264.31	552.6 ^c^	˂1
**30**	dihydro-4-hydroxy-2(3*H*)-furanone	1266.84	3741 ^c^	˂1
**38**	dipropyl disulfide	35.56	130 ^d^	˂1
**8**	2-furanmethanol	1119.00	4500 ^d^	˂1
**16**	benzaldehyde	131.79	750 ^d^	˂1
**45**	dimethyl sulfone	120.34	2158 ^c^	˂1
**15**	furfural	144.13	9562 ^d^	˂1
**1**	2-methyl-2-butanol	12.41	20,000 ^d^	˂1
**48**	pyrazine	41.12	300,000 ^d^	˂1
**28**	γ-butyrolactone	217.78	˃1000 ^d^	˂1

^a^ For the odor detective thresholds of compounds in water. ^b^ Data were from reference [44]. ^c^ Data were determined in this study. ^d^ Data were from reference [45].

**Table 4 molecules-26-07647-t004:** Concentration changes of potent odorants during storage in SP.

No.	Compound	Concentration (μg/kg) at Different Storage Time ^a^				
		1 Month	3 Months	10 Months	12 Months	14 Months
1	2-methyl-2-butanol	12.41 ± 0.07a ^b^	11.76 ± 0.13b	7.39 ± 0.08c	9.69 ± 0.1d	4.24 ± 0.04e
8	2-furanmethanol	1119 ± 8.09b	1501.91 ± 15.73a	237.55 ± 2.93d	128.12 ± 3.39e	1049.3 ± 13.85c
11	hexanal	553.03 ± 39.16e *^c^	1693.5 ± 16.6d *	4668.21 ± 62.38c *	18,618.34 ± 141.94b *	24,307.34 ± 52.02a *
12	3-methyl-2-butenal	13.37 ± 1.04c	8.63 ± 0.11cd	8.5 ± 0.25d	20.32 ± 4.47b	37.46 ± 0.16a
15	furfural	144.13 ± 0.05c	171.9 ± 0.68a	62.44 ± 0.44e	97.37 ± 0.63d	157.2 ± 2.91b
16	benzaldehyde	131.79 ± 0.29d	225.13 ± 0.42c	133.42 ± 2.47d	233.06 ± 1.62b	322.99 ± 0.48a
17	(*E*,*E*)-2,4-decadienal	164.46 ± 0.37b *	175.19 ± 1.11a *	55.54 ± 0.93e *	97.83 ± 0.89d *	160.07 ± 0.86c *
20	vanillin	34.72 ± 0.32e	95.86 ± 1.27c *	46.17 ± 0.49d	137.47 ± 2.85a *	132.16 ± 0.71b *
21	acetoin	177.9 ± 1.76d *	292.22 ± 16.33c *	114.4 ± 1.2e *	353.62 ± 2.93a *	337.72 ± 0.66b *
23	cyclopentenone	67.68 ± 0.7e	297.5 ± 0.53d	347.1 ± 12.21c	948.02 ± 5.22a	403.15 ± 14.95b
24	2(5*H*)-furanone	1642.24 ± 62.49b *	1996.08 ± 73.88a *	663.38 ± 10.61d	1262.45 ± 25.04c *	1902.13 ± 7.01a *
25	maltol	5073.26 ± 20.19d *	7433.15 ± 17.67c *	5072.21 ± 38.55d *	8839.38 ± 484.83b *	18,963.44 ± 119.53a *
28	γ-butyrolactone	217.78 ± 2.77d	387.77 ± 0.91b	216.32 ± 2.04d	329.89 ± 1.37c	443.5 ± 2.01a
30	dihydro-4-hydroxy-2(3*H*)-furanone	1244.31 ± 20.75b	2223.08 ± 77.41a	1043.69 ± 7.9c	686.75 ± 9.46d	1306.51 ± 11.56b
37	methyl propyl disulfide	28.9 ± 0.14d *	54.41 ± 0.25b *	24.16 ± 0.18e *	40.98 ± 0.01c *	102.15 ± 0.89a *
38	dipropyl disulfide	35.56 ± 0.19d	109.25 ± 1.51c	14.02 ± 0.2e	146.3 ± 0.28b *	282.7 ± 1.73a *
39	dimethyl trisulfide	29.77 ± 0.26c *	45.91 ± 0.46b *	17.45 ± 0.16e *	22.83 ± 0.08d *	69.42 ± 0.71a *
41	dimethyl sulfoxide	264.31 ± 2.5c	479.97 ± 0.92b	264.99 ± 2.58c	688.51 ± 7.7a *	686.37 ± 3.39a *
42	dipropyl trisulfide	9.92 ± 0.14d *	21.59 ± 0.41c *	4.5 ± 0.13e *	28.46 ± 0.44b *	99.08 ± 1.6a *
45	dimethyl sulfone	120.34 ± 0.71c	173.55 ± 3.55b	101.55 ± 4.38d	172.68 ± 11.89b	198.12 ± 1.14a
48	pyrazine	41.12 ± 0.1d	62.42 ± 1.86c	36.8 ± 0.25e	67.34 ± 0.38b	85.24 ± 0.41a
49	2-pentylfuran	19.65 ± 0.06d *	37.45 ± 0.35a *	8.69 ± 0.08e *	25.15 ± 1.12b *	23.39 ± 0.82c *
53	2-methylnaphthalene	16.93 ± 0.02b *	12.36 ± 0.14b *	13.5 ± 0.3c *	25.22 ± 0.02a *	11.03 ± 0.08e *

^a^ Concentration expressed as average concentration ± standard deviation (*n* = 3). ^b^ Different letters next to the values mean significant differences according to the Duncan’s multiple comparison tests (*p* < 0.05). ^c^“*” means OAV ≥ 1.

**Table 5 molecules-26-07647-t005:** OAV changes in potent odorants during storage in SP.

No.	Compound	OAV at Different Storage Time				
		1 Month	3 Months	10 Months	12 Months	14 Months
11	hexanal	230	706	1945	7758	10,128
17	(*E*,*E*)-2,4-decadienal	6091	6488	2057	3623	5929
20	vanillin	˂1	2	˂1	3	2
21	acetoin	13	21	8	25	24
24	2(5*H*)-furanone	2	3	˂1	2	3
25	maltol	24	35	24	42	90
37	methyl propyl disulfide	270	509	226	383	955
38	dipropyl disulfide	˂1	˂1	˂1	1	2
39	dimethyl trisulfide	3007	4638	1763	2306	7012
41	dimethyl sulfoxide	˂1	˂1	˂1	1	1
42	dipropyl trisulfide	52	114	24	150	521
49	2-pentylfuran	3	6	1	4	4
53	2-methylnaphthalene	6	4	5	8	4
	total	9699	12,515	6053	14,306	24,676

## Data Availability

All research data has been presented in this article and available from authors.

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
