# Peer review of "Characterization of Key Odorants in Scallion Pancake and Investigation on Their Changes during Storage"

_molecules, 2021, doi:10.3390/molecules26247647_

Round 1
Reviewer 1 Report
The authors present a work entitled "Characterization of Key Odorants in Scallion Pancake and Investigation on their Changes during Storage". The manuscript is well written and organized. The experiments are fully described and are presented correctly so that they can be easily reproduced. The results are presented in the form of tables and commented on in a timely manner with bibliographical references that are always relevant. In order to improve the general quality of the manuscript I suggest only the following minor changes: Authors should present the introduction in a richer way by adding more recent literature to better define the state of the art. The purpose of the work should be highlighted and enhanced more in the introduction. The conclusions should not only provide a brief summary of the results obtained, but also briefly illustrate the advantages and main novelties of the study, clearly highlighting the future prospects.
Author Response
The authors present a work entitled "Characterization of Key Odorants in Scallion Pancake and Investigation on their Changes during Storage". The manuscript is well written and organized. The experiments are fully described and are presented correctly so that they can be easily reproduced. The results are presented in the form of tables and commented on in a timely manner with bibliographical references that are always relevant. In order to improve the general quality of the manuscript I suggest only the following minor changes:
Comment 1 Authors should present the introduction in a richer way by adding more recent literature to better define the state of the art. The purpose of the work should be highlighted and enhanced more in the introduction.
Response 1 We have revised the introduction. If you are not satisfied with the revision, please tell us, and we will revise this part further, Thank you very much.
Comment 2 The conclusions should not only provide a brief summary of the results obtained, but also briefly illustrate the advantages and main novelties of the study, clearly highlighting the future prospects.
Response 2 The conclusions have been revised. Thank you.

Reviewer 2 Report
It’s very necessary for us to understand the key odorants of scallion pancake and their changes during storage, which may help improve food storage technology.
The comments and suggestions are as follow:
There is a grammatical mistake in a sentence (line 303): “Full-scan mode was use”.
The expression of “p<0.05” in the article should be italic as it presented at line 379.
Author Contributions should be revised carefully.
The format of references should be standardized.
Author Response
It’s very necessary for us to understand the key odorants of scallion pancake and their changes during storage, which may help improve food storage technology.
The comments and suggestions are as follow:
Comment 1 There is a grammatical mistake in a sentence (line 303): “Full-scan mode was use”.
Response 1 This sentence has been revised. Thank you.
Comment 2 The expression of “p<0.05” in the article should be italic as it presented at line 379.
Response 2 The format has been revised.
Comment 3 Author Contributions should be revised carefully.
Response 3 Author Contributions have been revised. Thank you.
Comment 4 The format of references should be standardized.
Response 4 The format of references have been revised.

Reviewer 3 Report
Comments to the Author
I have reviewed the manuscript entitled “Characterization of Key Odorants in Scallion Pancake and Investigation on their Changes during Storage”. The authors propose to characterize key odorants in scallion pancake (SP) and investigate on the changes of potent odorants during storage by sensomics. It is well structured, the bibliography is complete and up-to-date. Before accepting, there are a few deficiencies in this article need to be improved according to following list of comments:
General Comments:
The reconbinate model was baed on the result of quantification of key aroma compounds. Therefore, the section of 2.1 should be listed after the section of Quantitative analysis and determination of key odorants by OAVs. In addition, the identifications of (Z/E)-1-Propenyl propyl trisulfide and (Z/E)-3,6-Diethyl-1,2,4,5- tetrathiane was also baed on the result of GC-O, the section of 2.2 should be listed after the section of GC-O.
Specific Comments:
L.16: The “were” should be revised to “was”.
L.68-79: Regarding the quantitative descriptive analysis, ANOVA is essential. Please supplement the result of ANOVA.
L.75: The color of the period.
L.78-79: In Figure 1, the mean of “CSP” should be described in Figure caption.
L.199: In Table 3, authors should be listed the matrix of threshold of aroma compounds determined in literature. In addition, the matrix of SP might be very different from the matrix of threshold of aroma compounds reported. As a result, the result of OAV might be affected. Authors should be The author needs to point out this difference in the paper.
L.218-235: The section of “Changes of Aroma Compounds during Storage” should be added more discussion. In particular, the discussion of changes in aroma compounds was needed. In addition, in the preriod of storage, Is it possible to produce new key aroma compounds? More discussion should be added.
L.353-359: The description of the method of DA method was not detailed. The specific content of sensory group training needs to be listed.
Author Response
Reviewer 3
I have reviewed the manuscript entitled “Characterization of Key Odorants in Scallion Pancake and Investigation on their Changes during Storage”. The authors propose to characterize key odorants in scallion pancake (SP) and investigate on the changes of potent odorants during storage by sensomics. It is well structured, the bibliography is complete and up-to-date. Before accepting, there are a few deficiencies in this article need to be improved according to following list of comments:
General Comments:
Comment 1
The reconbinate model was baed on the result of quantification of key aroma compounds. Therefore, the section of 2.1 should be listed after the section of Quantitative analysis and determination of key odorants by OAVs. In addition, the identifications of (Z/E)-1-Propenyl propyl trisulfide and (Z/E)-3,6-Diethyl-1,2,4,5- tetrathiane was also baed on the result of GC-O, the section of 2.2 should be listed after the section of GC-O.
Response 1 The structure of this article has been adjusted. Thank you
Specific Comments:
Comment 2
L.16: The “were” should be revised to “was”.
Response 2 It has been revised. Thank you.
Comment 3 L.68-79: Regarding the quantitative descriptive analysis, ANOVA is essential. Please supplement the result of ANOVA.
Response 3 To the best of my knowledge, ANOVA is hardly performed to analysis the result of QDA among flavor analysis. I have listed the result of QDA below as mean value ± standard deviation. If there are any researches or references I should learn, please recommend me. Thank you.
|
Odor attribute |
sauce-like |
fatty |
sulfury |
sweet |
roasty |
green |
|
SP |
5.44±1.03 |
6.50±1.20 |
4.56±1.03 |
3.94±0.77 |
5.69±1.20 |
3.56±0.73 |
|
recombinate model |
4.31±1.08 |
6.75±1.29 |
4.00±1.10 |
2.56±0.89 |
5.25±1.18 |
4.06±1.09 |
Comment 4 L.75: The color of the period.
Response 4 The color of the period has been revised.
Comment 5 L.78-79: In Figure 1, the mean of “CSP” should be described in Figure caption.
Response 5 It should be SP; it has been revised in Figure . Thank you.
Comment 6 L.199: In Table 3, authors should be listed the matrix of threshold of aroma compounds determined in literature. In addition, the matrix of SP might be very different from the matrix of threshold of aroma compounds reported. As a result, the result of OAV might be affected. Authors should be The author needs to point out this difference in the paper.
Response 6 The detective threshold in water were applied to calculate the OAV in this manuscript. The bias caused by the thresholds from different matrix may affect the results of OAVs and finally affect the results of QDA for recombinate. Because the overall odor profiles of SP and recombinate are closer, we think the bias from the threshold is lower. We have supplied the relative content in the manuscript.
Comment 7 L.218-235: The section of “Changes of Aroma Compounds during Storage” should be added more discussion. In particular, the discussion of changes in aroma compounds was needed.
Response 7 Some discussion have been added.
Comment 8 In addition, in the preriod of storage, Is it possible to produce new key aroma compounds? More discussion should be added.
Response 8 I wouldn’t deny the possibility of identifying new aroma compounds in the period of storage. But, the study about changes of aroma compounds during storage was based on the potent compounds. These potent compounds may have an OAV closed to or greater than 1, which were of interest to us. At the same time, the OAVs of vanillin, dipropyl disulfide and dimethyl sulfoxide exceed 1 gradually during storage, which could be considered as the new key aroma compounds in the period of storage. Meanwhile, to find if some new key odorants were formed during storage, the odor profiles of these samples stored for some time were evaluated before the analyses; no new attributes were detected.
Comment 9 L.353-359: The description of the method of DA method was not detailed. The specific content of sensory group training needs to be listed.
Response 9 The paragraph has been revised as follow.
Six notes (sauce-like, fatty, sulfurous, sweet, roasty, green) were determined by preliminary experiment. For QDA, two sessions were performed. In the first session, the panelists were trained to distinguish six note attributes, and each of them was presented by one specific odorant solution, prepared by 50-fold to detective threshold in water, namely, sauce-like(dimethyl trisulfide), fatty(E,E-2,4-decdienal), sulfury(dipropyl trisulfide), sweet(maltol), roasty(pyrazine), green(hexanal). In the second session, the panelists were asked to score the intensity of each note on a scale of 0 to 9 integers, which 0 meant odorless and 9 meant very strong.
Round 2
Reviewer 3 Report
I have no more questions.